# Field and DNA-barcode based surveys reveal evidence of rare endemic fishes in the Rufiji River Basin

Jackson L. Saiperaki[1]*, Silvia F. Materu[1], Prisila A. Mkenda[1], Elly J. Ligate[1], Cyrus Rumisha[2]

1 Department of Biosciences, Sokoine University of Agriculture, Morogoro, Tanzania, 2 Department of Animal, Aquaculture and Range Sciences, Sokoine University of Agriculture, Morogoro, Tanzania

☯ These authors contributed equally to this work.
* jacksonlendoya89@gmail.com

**Data Availability Statement:** The data that support the finding of this study are available in NCBI databases under the accession number (OQ908874- OQ918545).

## Abstract

Endemic fish species have long supported the livelihoods of local communities in the Rufiji River Basin (RRB). However, destructive fishing practices have led to a concerning decline in endemic fish stocks. To assess these changes, this study employed key informant interviews, focus group discussions (FGDs), and fishery surveys to assess the historical and contemporary distribution of endemic fishes within the RRB. DNA barcoding was also used to verify species identities. Out of 37 reported fish species, 33 species (54.55% endemic and 45.45% exotic to RRB) were confirmed through DNA barcoding and morphological characteristics. About 5 species including, *Heterobranchus longifilis*, *Citharinus congicus*, *Labeo congoro*, *Mormyrus longirostris*, and *Labeobarbus leleupanus* were rarely found in the field, despite being classified as Least Concern by IUCN. Additionally, five species that were reported to be present in the RRB by experienced fishers were not captured during sampling. This highlights the need for validation of the existence of such species through eDNA metabarcoding. Moreover, due to the rarity of some species in the area, their IUCN assessment should be revisited.

## 1. Introduction

Freshwater fish have traditionally been a significant source of animal protein, income and employment to riparian communities globally [1]. In 2020 freshwater fisheries in Tanzania contributed for over 86% of total fish production and generated around two billion TZS [2]. However, unsustainable fishing practices driven by rapid population growth and high demand for fish protein [3] have resulted in a rapid decline of freshwater fish stocks, particularly in the Rufiji River Basin (RRB).

The decline in fish stocks in the RRB can be attributed to destructive fishing practices, such as poison fishing, dynamite fishing and the use of beach seine nets, as well as poor water quality from unsustainable agriculture and overgrazing [3, 4]. Additionally, the basin continues to shrink and the number of endemic fish species are declining due to land use change [5]. The

**Funding:** This study was funded by Sokoine University of Agriculture Research and Innovation Support (SUARIS) through the CoEF project (Conservation of Endemic Fishes project) under the grant number DPRTC/R/126/CoNAS/1/2022. The funder has no say on the design, data collection and analysis, writing of the manuscript and decision to submit a manuscript for publication consideration.

**Competing interests:** The author have declared that no competing interests exist.

native tribes of the RRB such as Ndamba and Pogoro have been engaged in fishing since time immemorial, but in the last 1–2 decades there is a great shift to crop farming as alternative source of food and livelihood support. Ndamba and Pogoro have a strong connection with endemic fish species, and thus their disappearance could have severe implications for household animal protein sources [6, 7].

Despite the designation of the Kilombero Valley Floodplain (KVFP) as a Ramsar site in 2002 and the establishment of the Nyerere National Park within the RRB [8, 9], the conservation of fish stocks in the RRB remains a critical issue. Unprotected areas within the RRB face significant fishing pressure, raising concerns about the potential disappearance of certain species from local catches [10]. Currently, the available information on the species composition in the region dates back over 20 years, originating from a study that identified 23 fish species in the RRB [11]. However, this study relied solely on morphological identification methods alone, which raise concerns about the potential existence of cryptic species and the possibility of misidentification [12]. Such inaccuracies can skew population assessments and conservation priorities, potentially leading to inadequate protection measures for vulnerable species. This lack of accurate and up-to-date information on species composition hinders conservation efforts, making it challenging to implement targeted interventions to protect vulnerable species and maintain ecosystem balance. Additionally, prevalent illegal fishing activities in the RRB [10] exacerbate these challenges, posing a direct threat to fish populations, especially rare and vulnerable species. Without accurate data on species composition and population dynamics, addressing and mitigating the impacts of illegal fishing activities become even more difficult. Hence, there is an urgent need to implement comprehensive monitoring programs that integrate advanced molecular techniques like DNA barcoding alongside traditional methods. This study integrated DNA barcoding and morphological identification techniques to reveal the composition of endemic fish species in the RRB. These approaches have been previously used in the country to uncover non-targeted tilapias among farmed fish and unveil protected elasmobranchs in Tanzanian fish markets [13, 14]. These approaches will provide more accurate assessments of species composition in the RRB, enabling better-informed conservation strategies and ensuring the long-term sustainability of its fish stocks.

## 2. Material and methods

### 2.1 Study site

The present study was conducted in the RRB which consists of the Kilombero River, the Great Ruaha, the Rufiji River and other small rivers [15]. Six landing sites in the RRB including Kidatu, Kivukoni, Mofu, Dinari, Ngalimila and Zombe were selected based on the availability and accessibility of landing sites (Fig 1). The RRB is the largest river basin in East Africa, which is rich in fish biodiversity [16]. It lies between 5.7˚ to 10.5˚S and 33.5˚ to 39˚E, covering an area of about 177,429 km2, which accounts for 20% of the total land area of Tanzania [17]. The RRB includes the KVFP, the largest seasonal freshwater lowland floodplain in East Africa [8]. It contains the Kilombero Valley Ramsar site (KVRS), an internationally recognized site of local and international importance. The Ramsar site covers an area of 796,735ha with the wetland catchment area of 40,000 $km^2$ [8]. The RRB also contains Nyerere National Park, which is the largest National Park in Africa covering an area of over 30,000 $km^2$ [9, 18]. The main economic activities in the RRB are fishing, crop production and livestock keeping [19]. The climatic condition of the RRB varies from tropical humid in the east to temperate in the southern highlands. In the east, the mean daily annual temperature is around 39˚C while it is around 23˚C in southern highlands [17]. The rainfall ranges from 250 mm in some areas to over 1800 mm on the east of the Udzungwa Mountain [17].

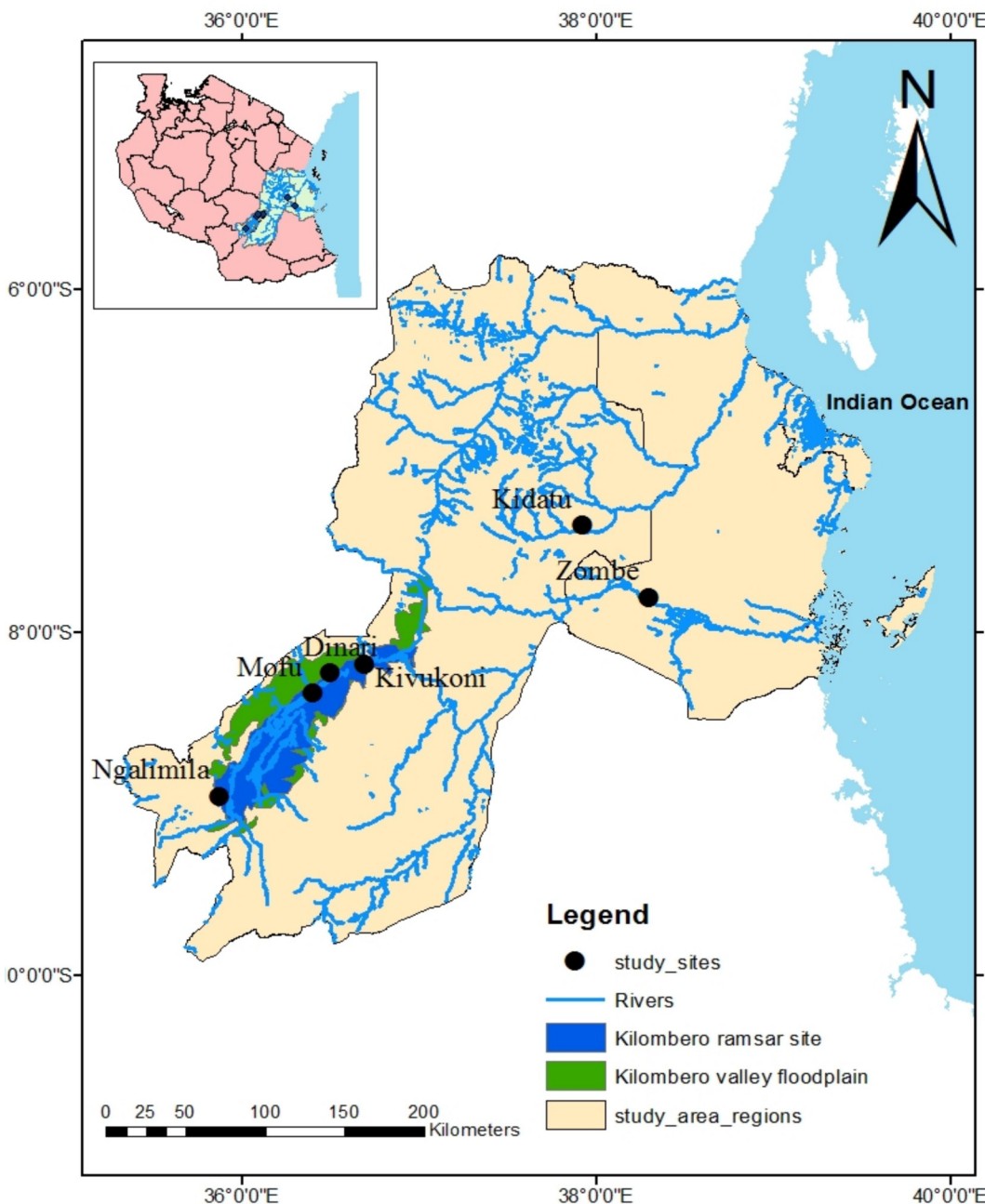

**Fig 1. Map showing the sampling sites in the Rufiji River Basin (RRB).** The map was created using ArcGIS software and shapefiles from National Bureau of Statistics (NBS) https://www.nbs.go.tz/index.php/en/census-surveys/gis/385-2012-phc-shapefiles-level-one-and-two. Accessed 10 April 2024.

## 2.2 Data collection

Fish sampling was conducted during two sampling seasons, between July 2022 (the onset of the dry season) and January 2023 (the onset of the wet season). A total of 46 different species were collected at six landing site of the RRB. Fish were initially identified using the available fish identification keys [20, 21]. Fish species that showed potential differences from those

**Table 1. Central coordinates and numbers of fins clip sampled within the Rufiji River Basin (RRB).**

| Site | Coordinates | | Number of fin clips sampled |
|---|---|---|---|
| | Latitudes (° S) | Longitudes (° E) | |
| **Rufiji River** | | | |
| Zombe | 7.80 | 38.30 | 2 |
| **Ruaha River** | | | |
| Kidatu | 7.38 | 37.92 | 2 |
| **Kilombero River** | | | |
| Kivukoni | 8.19 | 36.69 | 34 |
| Mofu | 8.36 | 36.40 | 1 |
| Dinari | 8.24 | 36.58 | 2 |
| Ngalimila | 8.96 | 35.87 | 5 |
| Total | | | 46 |

already sampled were specifically collected from each landing sites. For every landing site, where samples were taken (Table 1), coordinate points were recorded using Geographical Positioning System (GPS) device. Fin clip tissues of about 0.05 grams were cut from each fish, stored in 1.5 ml micro centrifuges and preserved using 99.9% ethanol until further analysis. Additionally, three focus group discussions were conducted to gather information about species composition, local fish identification techniques, fishing trends and fish management strategies. In-depth interviews were conducted to 4 groups of key informants including village elders, environmental management officers, fisheries officers and village chairpersons to gather information about the composition of fish in the RRB, local fish identification techniques.

## 2.3 Ethical statement

The fish sampled in this study were obtained from landing sites in the study area where they had already been caught by local fishers for human consumption. Therefore, no additional methods of sacrifice, anesthesia, or analgesia were required or administered by the researchers. The sampling process involving collecting fin clips from deceased fish only, ensuring that nor further suffering was inflicted. Authorizations for sampling were obtained from the Sokoine University of Agriculture and the Tanzania Ministry of Regional Administration and Local Government under permit number AB.307/323/01/24.

## 2.4 DNA extraction, COI amplification, and sequencing

Genomic DNA was extracted from each sample using the TIANamp Genomic DNA kit (TIANGEN Biotech, Beijing) according to the manufacturer's protocol. Then the quality of each DNA extract was evaluated on 1% agarose gel before further analysis [22]. Thereafter, fragments (620 base pairs) of the cytochrome oxidase subunit I gene (COI) were amplified from the DNA extracts of each sample in a T100$^{TM}$ Thermal cycler machine (Bio-Lab Inc, GA, USA) using the Forward primer FishFI (5′–TCAACCAACCACAAAGACATTGGCAC–3′) and the reverse primer FishR1 (5′–TAGACTTCTGGGTGGCCAAAGAATCA–3′) [23]. Amplification reactions were done in a total volume of 35 μL consisting of 2 μL template DNA, 1 x One-Taq 2X Master Mix with Standard Buffer (New England BioLabs Inc., MA, USA), 5 mg bovine serum albumin and 0.3 μM of each primer. Each reaction was initially denatured at 94°C for 5 min, followed by 35 cycles of 94°C for 40 s, 54°C for 45 s and 72°C for 60 s. The final extension of 72°C for 15 min was added to ensure complete elongation. The quality of each PCR product

was checked on a 1.5% agarose gels. The successful PCR amplicons were Sanger sequenced by Macrogen Europe Laboratory in the ABI 3730XL automated sequencer (Applied Bio systems, Foster City, USA).

## 2.5 Data analysis

A total of 46 samples were successfully analysed. The obtained sequences were edited to trim the ends and aligned using ClustalW algorithm as implemented in the program MEGA ver. 11 [24] to obtain sequences with equal length of 600 base pairs. Each sequence was then compared with COI sequences in the GenBank Nucleotide Database using the BLAST (Basic Local Alignment Search Tool) and BOLD (Barcode of Life Data System). The sequences were then submitted to GenBank and accession numbers (OQ908874- OQ918545) were provided. At least 90.91% of the unknown fish were identified to species level and the samples were classified to family, genus and species following the Linnaean taxonomy.

The Bayesian phylogenetic tree was constructed using BEAST ver 2.5 [25] to assess the evolutionary relationships among species. The analysis employed a relaxed uncorrelated log-normal molecular clock and a general time- reversible evolutionary model, running for 10 million generations. The tree was annotated using TreeAnnotator ver 1.10 and visualized using Fig-Tree ver 1.4. The COI sequence of Leopard whip ray *Himantura leoparda* with the accession number MK422130 was retrieved from GenBank and included in the dataset as outgroup.

## 3. Results

### 3.1 Fish diversity

Fishers and the key informants mentioned a total of 37 different fishes found within the RRB. About 5 fish species were not verified during fishery survey suggesting that they are either no longer abundant in the wild or they are present in a very low numbers (Table 2). About 5 species including *H. longifilis*, *C. congicus*, *L. congoro*, *M. longirostris*, and *L. leleupanus* were rarely found in the field. Moreover, fishers in the RRB used the local identification techniques such as fish morphology including the size of the fish and number or structure of fins to identify fish. This identification knowledge was obtained from village elders and the experienced fishermen. The provided local names, however, do not reflect the Linnaean taxonomy and the DNA barcoding results. For example, *Synodontis multipuctatus* was named as ngogo ng'andu and ngogo mwanajeshi while *Labeo congoro* was named as mtuku and ningu depending on morphological characteristics and stage of development. Additionally, one local name was given to more than one species, particularly those with similar morphologies. For example two different species of tilapia *Oreochromis korogwe* and *Oreochromis urolepis* were reported as perege, while *Glossogobius giuris* and *Eleotris klunzingerii* were reported as bubu mchanga while *Hippopotamyrus* spp. and *Petrocephalus affinis* were reported as ndipi (Table 2). Furthermore, although fishers could distinguish matured bula *Schilbe moebiusii* and luepe *Eutropiellus longifilis*, they could not distinguish juveniles of these species due to their similar morphologies.

### 3.2 Confirmation of morphologically identified species through DNA barcoding

A total of 46 COI barcode sequences representing 33 different species belonging to 24 different genera, 11 different families and 8 different orders were obtained from the sampled specimens. About 18 (54.55%) out of 33 species were endemic while 15 (45.45%) species were exotic to RRB (Fig 2). These endemic species included; njuju *Brycinus affinis*, mtuku or ningu *Labeo*

**Table 2. The local names of endemic fishes sampled in the Rufiji River Basin (RRB) and their corresponding Linnaean classification according to DNA barcode results.**

| No | Local name | Linnaean classification | | |
| --- | --- | --- | --- | --- |
| | | Family | Genus | Species |
| 1 | Benasongo | Cyprinidae | Enteromius | *Enteromius apleurogramma* |
| 2 | Bubu mchanga | Gobiidae | Glossogobius | *Glossogobius giuris* |
| 3 | Bubu mchanga | Eleotridae | Eleotris | *Eleotris klunzingerii* |
| 4 | Bula | Schilbeidae | Schilbe | *Schilbe moebiusii* |
| 5 | Gugutuu | Anabantidae | Ctenopoma | *Ctenopoma* spp |
| 6 | Jwalajwala | **Not verified** | | |
| 7 | Kambale | Clariidae | Clarias | *Clarias gariepinus* |
| 8 | Kibenamdenge | **Not verified** | | |
| 9 | Kitoga | Bagridae | Bagrus | *Bagrus orientalis* |
| 10 | Luepe | Schilbeidae | Eutropielus | *Eutropiellus longifilis* |
| 11 | Mbala | Citharinidae | Citharinus | *Citharinus congicus* |
| 12 | Mbewe | Alestidae | Brycinus | *Brycinus* spp |
| 13 | Mgundu | Alestidae | Alestes | *Alestes stuhlmanni* |
| 14 | Mjongwa | Clariidae | Heterobranchus | *Heterobranchus longifilis* |
| 15 | Mkunga | Anguillidae | Anguilla | *Anguilla bangelensis* |
| 16 | Mkuyu | Cyprinidae | Labeobarbus | *Labeobarbus leleupanus* |
| 17 | Mtuku | Cyprinidae | Labeo | *Labeo congoro* |
| 18 | Ndipi | Mormyridae | Petrocephalus | *Petrocephalus affinis* |
| 19 | Ndipi | Mormyridae | Hippopotamyrus | *Hippopotamyrus* spp |
| 20 | Ndipi kongwe | Mormyridae | Pollimyrus | *Pollimyrus nigrican* |
| 21 | Ndipi mdomo mfupi | Mormyridae | Marcusenius | *Marcusenius livingstonii* |
| 22 | Ndipi mdomo mrefu | Mormyridae | Marcusenius | *Marcusenius macrolepidatus* |
| 23 | Ndipi miera | **Not verified** | | |
| 24 | Ndipi namani | Mormyridae | Hippopotamyrus | *Hippopotamyrus grahami* |
| 25 | Ndungu | Distichodontidae | Distichodus | *Distichodus petersii* |
| 26 | Ngogo dongo | Mochokidae | Synodontis | *Synodontis multipuctatus* |
| 27 | Ngogo mwanajeshi | Mochokidae | Synodontis | *Synodontis multipuctatus* |
| 28 | Ngogo mweusi | Mochokidae | Synodontis | *Synodontis rukwaensis* |
| 29 | Ngogo ng'andu | Mochokidae | Synodontis | *Synodontis rufigiensis* |
| 30 | Ngolya | **Not verified** | | |
| 31 | Ngurufi | Cyprinidae | Labeo | *Labeo coubie* |
| 32 | Ningu | Cyprinidae | Labeo | *Labeo congoro* |
| 33 | Njege | Alestidae | Hydrocynus | *Hydrocynus tanzaniae* |
| 34 | Njuju | Alestidae | Brycinus | *Brycinus affinis* |
| 35 | Njuju mkapa | Alestidae | Brycinus | *Brycinus sadleri* |
| 36 | Perege 1 | Cichlidae | Oreochromis | *Oreochromis korogwe* |
| 37 | Perege 2 | Cichlidae | Oreochromis | *Oreochromis urolepis* |
| 38 | Sheta | Clariidae | Clarias | *Clarias werneri* |
| 39 | Sulusulu | Mormyridae | Mormyrus | *Mormyrus longirostris* |
| 40 | Sulusulu vihondi | **Not verified** | | |

*congoro*, bula *S. moebiusii*, ndipi mdomo mrefu *Marcusenius macrolepidatus*, ndungu *D. petersii*, njege *H. tanzaniae*, kitoga *B. orientalis*, ngogo ng'andu *Synodontis rufigiensis*, ngogo mweusi *Synodontis rukwaensis*, mbala *Citharinus congicus*, tilapia *O. urolepis*, benasongo *Enteromius apleurogramma*, mbewe *Brycinus* spp, luepe *E. longifilis*, mgundu *Alestes*

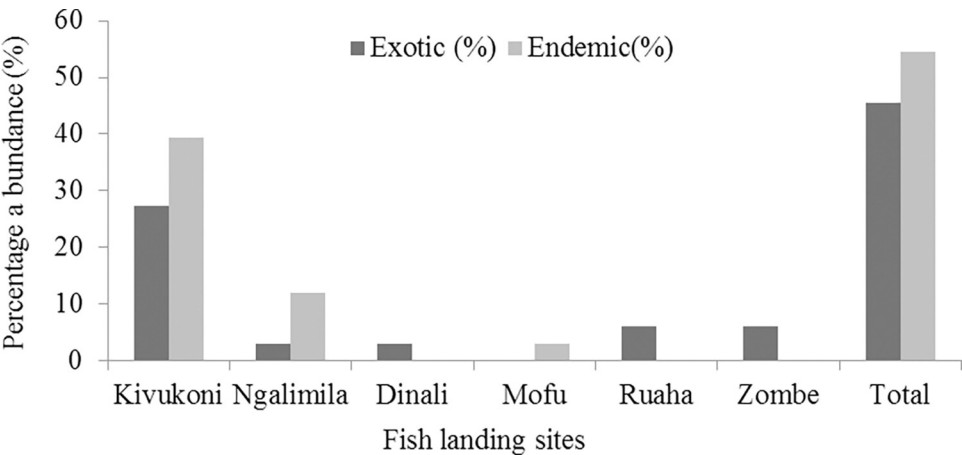

**Fig 2. Percentage of endemic and exotic fish sampled between July 2022 and January 2023 in six different landing sites within the Rufiji River Basin (RRB).**

*stuhlmanni*, ndipi *P. affinis*, ndipi *Hippopotamyrus* spp and ndipi mdomo mfupi *Marcusenius livingstonii*. About 13 different fish species were identified using GenBank and BOLD databases. However, higher identities (98.87%-99.84%) failed to confirm ndipi *P. affinis* and sulu-sulu *Mormyrus longirostris* while low identities confirmed mbala *C. congicus* (93.96%) and ndipi kongwe *Pollimyrus nigrican* (96.73%) in GenBank database (Table 3). The taxonomic identity of 20 different fish species were not confirmed using DNA barcode alone due to lack of reference barcodes in the GenBank and BOLD databases. Therefore, the integration of DNA barcode results and morphological identification was used to confirm the identity of the 20 fish species. Yet, the identification sheets were poor for ndipi *Hippopotamyrus* spp, mbewe *Brycinus* spp and gugutuu *Ctenopoma* spp.

### 3.3 Phylogenetic analysis of experimental fish species

The bayesian phylogenetic analysis performed from 46 nucleotide sequences (Fig 3) provided additional confirmation to the identified fish species. Closely related species were clustered under the same node implying that the amplified barcodes correctly identified the species.

### 3.4 Conservation status

It was revealed that 90.91% (30 different fish species) of the identified species are categorized by IUCN as least concern (LC), 3.03% as near threatened (NT), and 6.06% as vulnerable (VU) (Table 3). Hence, none of the sampled fish species is either endangered or critically endangered. Similarly, none of the sampled fish species is either CITES protected or protected by Tanzanian laws.

## 4. Discussion

The present study revealed 33 different fish species in the RRB. This number is higher than the number reported in a previous study [11] which showed that there was only 23 different fish species. The variation in results can be attributed to differences in sampling techniques employed, limited sampling sites and shorter duration of sampling. Therefore, a total of 10 fish species identified in this study were not reported in the earlier studies. These newly identified species include, bubu mchanga *G. giuris*, *E. klunzingerii*, gugutuu *Ctenopoma* spp, luepe *E. longifilis*, mkuyu *L. leleupanus*, ndipi *P. affinis*, ndipi kongwe *P. nigrican*, ngogo mwanajeshi *S.*

**Table 3. Fish species identification from GenBank and BOLD databases, conservation status and the number of samples obtained from fish species sampled between July 2022 and January 2023 in the Rufiji River Basin (RRB).**

| Scientific name | Accession no. | GenBank Species name | Identity (%) | BOLD Species name | Identity (%) | Number of samples | IUCN Red list category |
|---|---|---|---|---|---|---|---|
| *Brycinus affinis* | OQ908874 | **Alestes spp** | **92.57** | **No match** | **0** | 1 | LC |
| *Labeo congoro* | OQ908912 OQ908894 | **Labeo lineatus** | **96.89** | **Labeo lineatus** | **97.40** | 2 | LC |
| *Schilbe moebiusii* | OQ908911 | **Schilbe intemedius** | **93.98** | **No match** | **0** | 1 | LC |
| *Marcusenius macrolepidatus* | OQ908903 OQ908875 | **Campylomormyrus numenius** | **93.27** | **Marcusenius livingstonii** | **98.34** | 2 | LC |
| *Oreochromis korogwe* | OQ915200 OQ915199 | *Oreochromis korogwe* | 99.66 | *Oreochromis korogwe* | 99–100 | 2 | LC |
| *Distichodus petersii* | OQ908900 | *Distichodus petersii* | 98.04 | *Distichodus petersii* | 98.16 | 1 | VU |
| *Hydrocynus tanzaniae* | OQ918545 | **Hydrocynus vittatus** | 94.92 | **No match** | **0** | 1 | LC |
| *Clarias gariepinus* | OQ908902 | *Clarias gariepinus* | 99.84 | *Clarias gariepinus* | 99.84 | 1 | LC |
| *Bagrus orientalis* | OQ908901 | **Bagrus caeruleus** | **93.13** | **No match** | **0** | 1 | LC |
| *Synodontis rufigiensis* | OQ908910 OQ908889 | **Synodontis spp** | **94.96** | **No match** | **0** | 2 | LC |
| *Synodontis rukwaensis* | OQ908899 OQ908888 | *Synodontis rukwaensis* | 99.38 | *Synodontis rukwaensis* | 99.67 | 2 | LC |
| *Citharinus congicus* | OQ908898 OQ908881 | *Citharinus congicus* | 93.96 | **No match** | 0 | 2 | LC |
| *Anguilla bengalensis* | OQ908897 OQ908893 OQ908909 | *Anguilla bengalensis* | 100 | *Anguilla bengalensis* | 100 | 3 | NT |
| *Labeo coubie* | OQ918544 | **Labeo forskalii** | **95.57** | **No match** | **0** | 1 | LC |
| *Hippopotamyrus grahami* | OQ915201 | **Pollimyrus isidori** | **90.15** | **No match** | **0** | 1 | LC |
| *Hippopotamyrus* spp | OQ915198 | **Ciphomyrus discorhynchus** | **96.20** | **No match** | **0** | 1 | LC |
| *Mormyrus longirostris* | OQ908896 OQ908879 | **Mormyrus rume** | **98.87** | **Mormyrus tapirus** | **98.71** | 2 | LC |
| *Oreochromis urolepis* | OQ908895 | **Oreochromis korogwe** | **97.93** | *Oreochromis urolepis* | 98.36 | 1 | LC |
| *Pollimyrus nigrican* | OQ908892 | *Pollimyrus nigrican* | 96.73 | **No match** | **0** | 1 | LC |
| *Enteromius apleurogramma* | OQ908908 OQ908907 | *Enteromius apleurogramma* | 100 | *Enteromius apleurogramma* | 100 | 2 | LC |
| *Brycinus sadleri* | OQ908877 | **Brycinus sp.epulu** | 97.03 | **Brycinus lateralis** | **98.69** | 1 | LC |
| *Synodontis multipuctatus* | OQ908891 OQ908890 | **Synodontis victoriae** | 96.08 | **Synodontis victoriae** | **97.07** | 2 | LC |
| *Clarias werneri* | OQ908887 | **Clarias alluaudi** | **95.94** | **No match** | **0** | 1 | LC |
| *Brycinus* spp | OQ908886 | **Brycinus nurse** | **91.31** | **No match** | **0** | 1 | LC |
| *Eutropielus longifilis* | OQ908906 OQ908885 OQ908884 | **Schilbe intemedius** | **91.06** | **No match** | **0** | 3 | LC |
| *Labeobarbus leleupanus* | OQ908883 | **Labeobarbus robertsi** | **96.27** | **No match** | **0** | 1 | VU |
| *Heterobranchus longifilis* | OQ908882 | *Heterobranchus longifilis* | 98.72 | **Heterobranchus spp** | **98.86** | 1 | LC |
| *Alestes stuhlmanni* | OQ908880 | **Alestes baremoze** | **91.58** | **No match** | **0** | 1 | LC |
| *Petrocephalus affinis* | OQ908878 | **Petrocephalus catostoma** | **99.84** | **Petrocephalus catostoma** | **97.18** | 1 | LC |
| *Ctenopoma* spp | OQ908876 | **Ctenopoma muriei** | **93.81** | **No match** | **0** | 1 | LC |
| *Glossogobius giuris* | OQ918543 | *Glossogobius giuris* | 98.35 | *Glossogobius giuris* | 98.8 | 1 | LC |
| *Eleotris klunzingerii* | OQ908905 | *Eleotris klunzingerii* | 100 | *Eleotris klunzingerii* | 100 | 1 | LC |
| *Marcusenius livingstonii* | OQ908904 | **Ciphomyrus discorhynchus** | **93.23** | **No match** | **0** | 1 | LC |

LC = least concern, VU = vulnerable and NT = n.

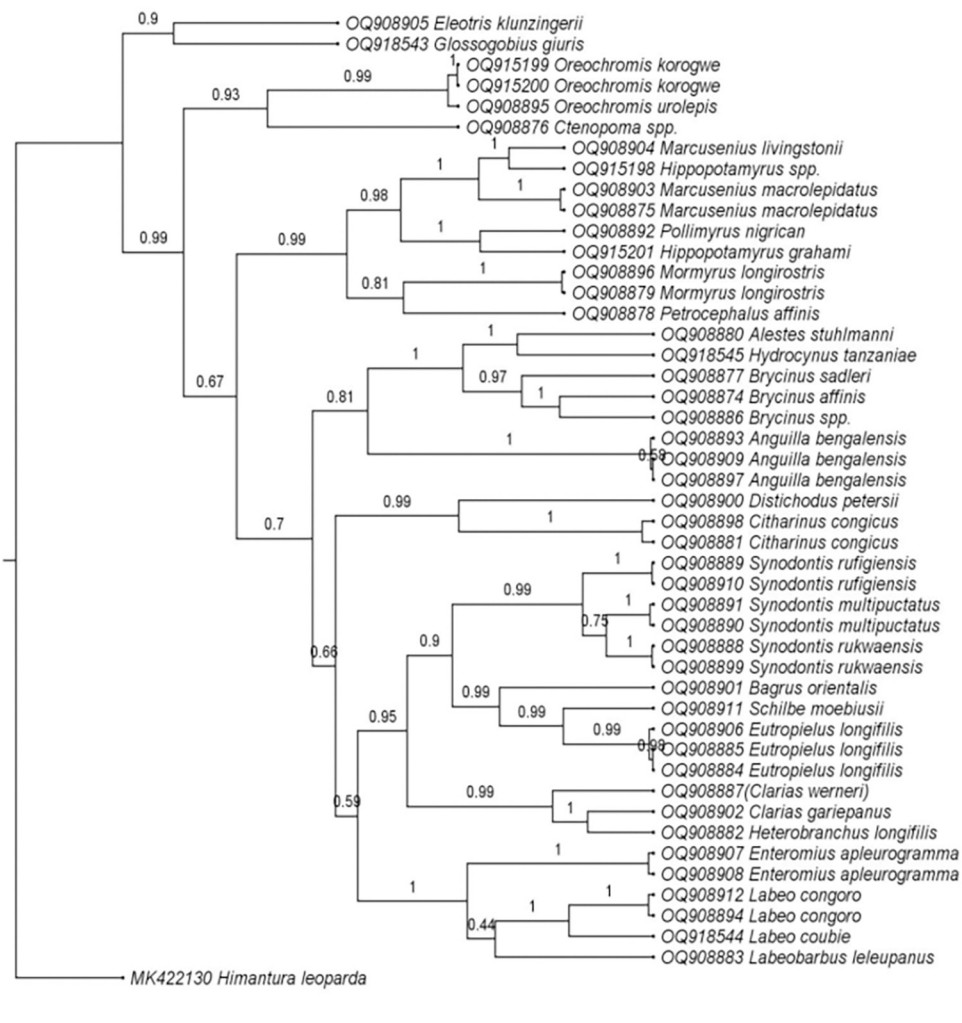

**Fig 3. Bayesian phylogenetic tree showing clustering patterns of cytochrome oxidase subunit I sequences of endemic and exotic fish species sampled in the Rufiji River Basin (RRB) between July 2022 and January 2023.**

*multipuctatus*, ngogo ng'andu *S. rufigiensis* and ndipi mdomo mrefu *M. macrolepidatus*. Eighteen out of 33 species were endemic to RRB while 15 were exotic. The presence of a high number of exotic fish species poses a serious threat to the endemic fish populations. Some of these exotic species can act as competitors, predators, or even hybridize with the endemic species, further exacerbating the risk of extinction [26]. The present study also confirmed the presence of *H. longifilis*, *C. congicus* and *L. coubie* contrary to study conducted by [27] which revealed that the species have disappeared in the RRB. However, the fact that these species were rare in the catch suggests that the current IUCN assessment of them as Least Concern should be revisited. This is particularly critical for *H. longifilis* because it was found at only one site and was reported by experienced fishers to be among the fishes that were highly abundant in the past but are currently rare.

The local fish identification techniques used was found to be inaccurate, leading to numerous contradictions, especially when distinguishing closely related species. Despite using the Field guide for freshwater fishes of Tanzania [20], there were limitations in the identification sheets, particularly for certain fish species. This is similar to the study conducted in the study

area [11] which showed the limitation of the identification sheets in identifying Mbewe *Brycinus* spp and Sheta *C. werneri*.

DNA barcoding alone confirmed identities of 13 species. However, low identities were used to confirm some species in the GenBank database while higher identities failed to confirm the identity of ndipi *P. affinis* and sulusulu *M. longirostris*, suggesting a high probability of tentative, incorrect or low-quality sequences being submitted to the database [28]. BOLD database confirmed less species than GenBank. However, most of the confirmed species were identified with 99–100% identities. This reveals that BOLD database has greater resolution than GenBank database [29]. The COI sequences of 21 fish species have not been recorded in the GenBank database, and the COI sequences of 17 fish species do not match any sequence in the BOLD database. Thus, this study added COI sequences for 21 fish species to the GenBank database and introduced sequences for 17 fish species that did not previously exist in the BOLD database. Furthermore, fish species identified from this study would help to solve the problem of unidentified species from the previous studies [11, 27]. Some fish species were however, not verified through DNA barcoding alone due to absence of corresponding COI sequences in the GenBank and BOLD Database. The integration of DNA sequencing information with the morphological traits of the fish showed great efficiency.

The constructed phylogenetic tree provided similar classification concerning taxonomy and morphological traits of the fishes. All closely related species were clustered under the same nodes revealing that the amplified barcodes correctly identified the species. The results of the present study indicate that none of the sampled fish in the RRB are classified as endangered or critically endangered according to the IUCN. However, due to the rarity of some species in the catch, their IUCN assessment should be revisited. This is critical for species such as mjongwa *H. longifilis*, mbala *C. congicus*, ningu *L. congoro*, sulusulu *M. longirostris*, and mkuyu *L. leleupanus* because they were particularly rare. These rare species require reassessment and reclassification as their current IUCN criteria does not accurately reflect their actual status on the ground. Additionally, because none of the rare species are listed in either CITES Appendices or the Third Schedule of the Tanzania Fisheries (Amendment) Regulations of 2009. This implies that there are currently no specific legal measures in place to regulate or protect these fish species from overexploitation or illegal trade. This highlights the need to update CITES Appendices and the Third Schedule of the Tanzania Fisheries (Amendment) Regulations of 2009 to include the above-mentioned rare species if they are to be protected from extinction. Furthermore, the absence of some reported species during sampling does not conclusively indicate their complete disappearance in the RRB; instead, it calls for further studies employing environmental DNA (eDNA) to confirm the presence of these species.

## 5. Conclusion

The present study confirmed 33 different species in the RRB, including species that were reported to have disappeared. However, some species were rarely found in the field despite being classified as Least Concern by the IUCN, suggesting the need for their IUCN Red List status to be reevaluated. Additionally, the presence of rare species suggests the need to protect them in the RRB to prevent further decline in fish populations. This can be achieved through promoting sustainable fishing practices by raising awareness among local fishers about techniques that minimize harm to fish populations and their habitats. Furthermore, the expansion of protected areas within the RRB could provide safe havens for rare species, potentially reversing the observed declining trends. Moreover, the findings of this study should be validated using environmental DNA (eDNA) to confirm the existence of species reported to have disappeared.

## Acknowledgments

We would like to express our gratitude to the local community in the RRB who helped us during fieldwork. We extend our appreciation to the Tanzanian Ministry of Regional Administration and Local Government for issuing the necessary permits. Lastly, we acknowledge the contributions of the reviewers, whose constructive criticism greatly improved this work.

## Author Contributions

**Conceptualization:** Jackson L. Saiperaki, Silvia F. Materu, Elly J. Ligate, Cyrus Rumisha.

**Data curation:** Jackson L. Saiperaki, Silvia F. Materu, Prisila A. Mkenda, Cyrus Rumisha.

**Formal analysis:** Prisila A. Mkenda.

**Funding acquisition:** Prisila A. Mkenda.

**Investigation:** Jackson L. Saiperaki, Elly J. Ligate.

**Methodology:** Jackson L. Saiperaki, Silvia F. Materu, Cyrus Rumisha.

**Project administration:** Elly J. Ligate.

**Resources:** Cyrus Rumisha.

**Software:** Jackson L. Saiperaki, Cyrus Rumisha.

**Supervision:** Silvia F. Materu, Cyrus Rumisha.

**Validation:** Jackson L. Saiperaki, Silvia F. Materu.

**Visualization:** Jackson L. Saiperaki, Prisila A. Mkenda.

**Writing – original draft:** Jackson L. Saiperaki, Silvia F. Materu, Prisila A. Mkenda, Elly J. Ligate, Cyrus Rumisha.

**Writing – review & editing:** Jackson L. Saiperaki, Silvia F. Materu, Elly J. Ligate, Cyrus Rumisha.

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
