## [Decision Letter · Decision Letter 0]

10 May 2024

PONE-D-24-11523Field and DNA-barcode based surveys reveal evidence of rare endemic fishes in the Rufiji River Basin, Tanzania.PLOS ONE

Dear Dr. Saiperaki,

Thank you for submitting your manuscript to PLOS ONE. After careful consideration, we feel that it has merit but does not fully meet PLOS ONE’s publication criteria as it currently stands. Therefore, we invite you to submit a revised version of the manuscript that addresses the points raised during the review process. Please submit your revised manuscript by Jun 24 2024 11:59PM. If you will need more time than this to complete your revisions, please reply to this message or contact the journal office at plosone@plos.org. Please include the following items when submitting your revised manuscript:A rebuttal letter that responds to each point raised by the academic editor and reviewer(s). You should upload this letter as a separate file labeled 'Response to Reviewers'.A marked-up copy of your manuscript that highlights changes made to the original version. You should upload this as a separate file labeled 'Revised Manuscript with Track Changes'.An unmarked version of your revised paper without tracked changes. You should upload this as a separate file labeled 'Manuscript'.

We look forward to receiving your revised manuscript.

Kind regards,

Feng ZHANG, Ph.D.

Academic Editor

PLOS ONE

Journal Requirements:

" This study was funded by Sokoine University of Agriculture Research and Innovation Support (SUARIS) through the CoEF project (Conservation of Endemic Fishes project) under the grant number DPRTC/R/126/CoNAS/1/2022. The funder has no say on the design, data collection and analysis, writing of the manuscript and decision to submit a manuscript for publication consideration."

4. We note that [Figure 1] in your submission contain [map/satellite] images which may be copyrighted. All PLOS content is published under the Creative Commons Attribution License (CC BY 4.0), which means that the manuscript, images, and Supporting Information files will be freely available online, and any third party is permitted to access, download, copy, distribute, and use these materials in any way, even commercially, with proper attribution. For these reasons, we cannot publish previously copyrighted maps or satellite images created using proprietary data, such as Google software (Google Maps, Street View, and Earth). For more information, see our copyright guidelines: http://journals.plos.org/plosone/s/licenses-and-copyright.

Reviewers' comments:

Reviewer's Responses to Questions

**Comments to the Author**

1. Is the manuscript technically sound, and do the data support the conclusions?

Reviewer #1: Partly

2. Has the statistical analysis been performed appropriately and rigorously? 

Reviewer #1: No

3. Have the authors made all data underlying the findings in their manuscript fully available?

Reviewer #1: Yes

4. Is the manuscript presented in an intelligible fashion and written in standard English?

Reviewer #1: Yes

5. Review Comments to the Author

Reviewer #1: The research is well-designed however, there are a few points that need clarification in the text, and the phylogenetic tree needs to be reconstructed using different approaches.

Major Revisions:

- Table 3: Species such as Labeo congoro, Marcusenius macrolepidatus, Mormyrus longirostris, Oreochromis urolepis, Brycinus sadleri, and Petrocephalus affinis showed over 97% coverage with other species during the NCBI or BOLD blast analysis. Do the authors have a plan for these species? Were these species uploaded to BOLD under their scientific names as given by the authors, or according to their coverage from the blast analyses?

- 2.5 Data Analysis: "‘The evolutionary history was inferred by using the maximum likelihood method and Tamura-Nei model. The bootstrap consensus tree inferred from 500 replicates was used to analyse the evolutionary history of the identified species. Branches corresponding to partitions reproduced in less than 75% bootstrap replicates were collapsed. The percentage of replicate trees in which the associated taxa clustered together in the bootstrap test 500 replicates was shown next to the branches.’

In the discussion section, it was also noted that all closely related species were clustered under the same nodes, indicating that the amplified barcodes correctly identified the species. However, there were instances where species from different orders and families clustered together, suggesting potential similarities or evolutionary relationships despite their taxonomic differences.

However, if the Himantura leoparda species (an outgroup) is positioned under the same node despite having a completely different background, this suggests that the tree may be unsuitable for this type of data. Therefore, I recommend that the authors use a different approach for reconstructing the tree, such as a Bayesian algorithm.

- Pictures of these species need to be given as supplementary data

- There should be barcoding gap analysis

Minor Revisions:

- 2.3 Ethical Statement: There is inconsistent spacing between lines.

- 3.1 Species Names: The names H. longifilis and C. congicus are not italicized.

- Figures: The resolution of the figures is very low.

6. PLOS authors have the option to publish the peer review history of their article (what does this mean?). If published, this will include your full peer review and any attached files.

Reviewer #1: **Yes: **ARZU KARAHAN

---

## [Author Response · Author response to Decision Letter 0]

11 Jun 2024

EDITOR COMMENTS

1. Please ensure that your manuscript meets PLOS ONE's style requirements, including those for file naming

Response: The manuscript was checked and formatted to make sure that it meets PLOS ONE’s style.

Response: The ethical statement explaining the permission to collect fin clips samples was included in Methods section. See line 102-105.

"This study was funded by Sokoine University of Agriculture Research and Innovation Support (SUARIS) through the CoEF project (Conservation of Endemic Fishes project) under the grant number DPRTC/R/126/CoNAS/1/2022. The funder has no say on the design, data collection and analysis, writing of the manuscript and decision to submit a manuscript for publication consideration. "Please state what role the funders took in the study. If the funders had no role, please state: "The funders had no role in study design, data collection and analysis, decision to publish, or preparation of the manuscript." 

If this statement is not correct you must amend it as needed. Please include this amended Role of Funder statement in your cover letter; we will change the online submission form on your behalf.

Response: The funder had no role in the study design, data collection and analysis, decision to publish, or preparation of the manuscript.

4. We note that [Figure 1] in your submission contain [map/satellite] images which may be copyrighted. All PLOS content is published under the Creative Commons Attribution License (CC BY 4.0), which means that the manuscript, images, and Supporting Information files will be freely available online, and any third party is permitted to access, download, copy, distribute, and use these materials in any way, even commercially, with proper attribution. For these reasons, we cannot publish previously copyrighted maps or satellite images created using proprietary data, such as Google software (Google Maps, Street View, and Earth). For more information, see our copyright guidelines: http://journals.plos.org/plosone/s/licenses-and-copyright. 

Response: The figure is replaced with the new one to avoid submitting the copyrighted images. The new map was created using ArcGIS software and shapefiles from National Bureau of Statistics (NBS) https://www.nbs.go.tz/index.php/en/census-surveys/gis/385-2012-phc-shapefiles-level-one-and-two. Accessed 10 April 2024.

REVIEWERS COMMENTS

Major Revisions

Reviewer #1: The research is well-designed however, there are a few points that need clarification in the text, and the phylogenetic tree needs to be reconstructed using different approaches.

Response: The phylogenetic tree was constructed using BEAST version 2.5, followed by annotation using TreeAnnotator version 1.10 and visualization using FigTree version 1.4. Notably, the tree exhibited a clustering pattern wherein closely related species were grouped together, indicating the accuracy of species identification through the amplified barcode. Consequently, additional clarifications were incorporated into the text, as outlined from line 193 to 209, to elucidate this observation.

Table 3: Species such as Labeo congoro, Marcusenius macrolepidatus, Mormyrus longirostris, Oreochromis urolepis, Brycinus sadleri, and Petrocephalus affinis showed over 97% coverage with other species during the NCBI or BOLD blast analysis. Do the authors have a plan for these species? Were these species uploaded to BOLD under their scientific names as given by the authors, or according to their coverage from the blast analyses?

Response: The highest similarities observed between these species with other species in NCBI or BOLD databases provide an alarm that there is high possibility of incorrect or low quality sequences that have been submitted to these databases. This suspicion is further underscored by the disparity between the morphological traits of the species documented in the database and those observed during our field surveys. Compounding the issue, the sequences of these species, identified by their scientific names, had not been previously uploaded to either BOLD or NCBI databases. In response, we uploaded these sequences in NCBI database for reference under the accession number OQ908874- OQ918545. However, we remain open to explore much on these species if funds were provided.

Data Analysis: "‘The evolutionary history was inferred by using the maximum likelihood method and Tamura-Nei model. The bootstrap consensus tree inferred from 500 replicates was used to analyse the evolutionary history of the identified species. Branches corresponding to partitions reproduced in less than 75% bootstrap replicates were collapsed. The percentage of replicate trees in which the associated taxa clustered together in the bootstrap test 500 replicates was shown next to the branches.’

In the discussion section, it was also noted that all closely related species were clustered under the same nodes, indicating that the amplified barcodes correctly identified the species. However, there were instances where species from different orders and families clustered together, suggesting potential similarities or evolutionary relationships despite their taxonomic differences.

However, if the Himantura leoparda species (an outgroup) is positioned under the same node despite having a completely different background, this suggests that the tree may be unsuitable for this type of data. Therefore, I recommend that the authors use a different approach for reconstructing the tree, such as a Bayesian algorithm.

Response: The bayesian phylogenetic analysis was performed as suggested by the reviewer. The tree clearly separated outgroup (Himantura leoparda) from other species and grouped closely related species under the same cluster. This clustering confirms that the amplified barcodes effectively identified the species.

Minor Revisions:

- 2.3 Ethical Statement: There is inconsistent spacing between lines.

Response: The ethical statement was checked as suggested by the reviewer to remove unnecessary spaces between lines

- 3.1 Species Names: The names H. longifilis and C. congicus are not italicized.

Response: The names were italicized see line 153

- Figures: The resolution of the figures is very low.

Response: The resolution of the figures was improved to 300dpi

---

## [Decision Letter · Decision Letter 1]

29 Aug 2024

Field and DNA-barcode based surveys reveal evidence of rare endemic fishes in the Rufiji River Basin, Tanzania.

PONE-D-24-11523R1

Dear Dr. Saiperaki,

We’re pleased to inform you that your manuscript has been judged scientifically suitable for publication and will be formally accepted for publication once it meets all outstanding technical requirements.

Kind regards,

Feng ZHANG, Ph.D.

Academic Editor

PLOS ONE

Additional Editor Comments (optional):

Reviewers' comments:

Reviewer's Responses to Questions

**Comments to the Author**

1. If the authors have adequately addressed your comments raised in a previous round of review and you feel that this manuscript is now acceptable for publication, you may indicate that here to bypass the “Comments to the Author” section, enter your conflict of interest statement in the “Confidential to Editor” section, and submit your "Accept" recommendation.

Reviewer #1: All comments have been addressed

2. Is the manuscript technically sound, and do the data support the conclusions?

Reviewer #1: Yes

3. Has the statistical analysis been performed appropriately and rigorously? 

Reviewer #1: Yes

4. Have the authors made all data underlying the findings in their manuscript fully available?

Reviewer #1: Yes

5. Is the manuscript presented in an intelligible fashion and written in standard English?

Reviewer #1: Yes

6. Review Comments to the Author

Reviewer #1: please do the below corrections;

1) Line 206: The title '3.3 Phylogenetic analysis of experimental fish species' should be simplified as '3.3 Phylogenetic analysis.'

2) The font size of the scale in Figure 3 needs to be increased.

7. PLOS authors have the option to publish the peer review history of their article (what does this mean?). If published, this will include your full peer review and any attached files.

Reviewer #1: **Yes: **Arzu Karahan

---

## [Editor Report · Acceptance letter]

10 Sep 2024

PONE-D-24-11523R1 

PLOS ONE

Dear Dr. Saiperaki, 

I'm pleased to inform you that your manuscript has been deemed suitable for publication in PLOS ONE. Congratulations! Your manuscript is now being handed over to our production team.

Kind regards, 

on behalf of

Dr. Feng ZHANG 

Academic Editor

PLOS ONE